# BanditPAM: Almost Linear Time $k$-Medoids Clustering via Multi-Armed Bandits

**Mo Tiwari**
Department of Computer Science
Stanford University
motiwari@stanford.edu

**Martin Jinye Zhang**
Department of Epidemiology
Harvard T.H. Chan School of Public Health
jinyezhang@hsph.harvard.edu

**James Mayclin**
Department of Computer Science
Stanford University
jmayclin@stanford.edu

**Sebastian Thrun**
Department of Computer Science
Stanford University
thrun@stanford.edu

**Chris Piech**
Department of Computer Science
Stanford University
piech@cs.stanford.edu

**Ilan Shomorony**
Electrical and Computer Engineering
University of Illinois at Urbana-Champaign
ilans@illinois.edu

## Abstract

Clustering is a ubiquitous task in data science. Compared to the commonly used $k$-means clustering, $k$-medoids clustering requires the cluster centers to be actual data points and supports arbitrary distance metrics, which permits greater interpretability and the clustering of structured objects. Current state-of-the-art $k$-medoids clustering algorithms, such as Partitioning Around Medoids (PAM), are iterative and are quadratic in the dataset size $n$ for each iteration, being prohibitively expensive for large datasets. We propose BanditPAM, a randomized algorithm inspired by techniques from multi-armed bandits, that reduces the complexity of each PAM iteration from $O(n^2)$ to $O(n \log n)$ and returns the same results with high probability, under assumptions on the data that often hold in practice. As such, BanditPAM matches state-of-the-art clustering loss while reaching solutions much faster. We empirically validate our results on several large real-world datasets, including a coding exercise submissions dataset from Code.org, the 10x Genomics 68k PBMC single-cell RNA sequencing dataset, and the MNIST handwritten digits dataset. In these experiments, we observe that BanditPAM returns the same results as state-of-the-art PAM-like algorithms up to 4x faster while performing up to 200x fewer distance computations. The improvements demonstrated by BanditPAM enable $k$-medoids clustering on a wide range of applications, including identifying cell types in large-scale single-cell data and providing scalable feedback for students learning computer science online. We also release highly optimized Python and C++ implementations of our algorithm[1].

## 1 Introduction

Many modern data science applications require the clustering of very-large-scale data. Due to its computational efficiency, the $k$-means clustering algorithm [32, 30] is one of the most widely-used

clustering algorithms. $k$-means alternates between assigning points to their nearest cluster centers and recomputing those centers. Central to its success is the specific choice of the cluster center: for a set of points, $k$-means defines the cluster center as the point with the smallest average *squared Euclidean distance* to all other points in the set. Under such a definition, the cluster center is the arithmetic mean of the cluster's points and can be computed efficiently.

While commonly used in practice, $k$-means clustering suffers from several drawbacks. Firstly, while one can efficiently compute the cluster centers under squared Euclidean distance, it is not straightforward to generalize to other distance metrics [40, 16, 8]. However, different distance metrics may be desirable in other applications. For example, $l_1$ and cosine distance are often used in sparse data, such as in recommendation systems [28] and single-cell RNA-seq analysis [38]; additional examples include string edit distance in text data [34], and graph metrics in social network data [33]. Secondly, the cluster center in $k$-means clustering is in general not a point in the dataset and may not be interpretable in many applications. This is especially problematic when the data is structured, such as parse trees in context-free grammars, sparse data in recommendation systems [28], or images in computer vision where the mean image is visually random noise [28].

Alternatively, $k$-medoids clustering algorithms [20, 21] use *medoids* to define the cluster center for a set of points, where for a set and an arbitrary distance function, the medoid is the point *in the set* that minimizes the average distance to all the other points. Mathematically, for $n$ data points $\mathcal{X} = \{x_1, \cdots, x_n\}$ and a given distance function $d(\cdot, \cdot)$, the $k$-medoids problem is to find a set of $k$ medoids $\mathcal{M} = \{m_1, \cdots, m_k\} \subset \mathcal{X}$ to minimize the overall distance of points to their closest medoids:

$$L(\mathcal{M}) = \sum_{i=1}^{n} \min_{m \in \mathcal{M}} d(m, x_i) \tag{1}$$

Note that the distance function can be arbitrary; indeed, it need not be a distance metric at all and could be an asymmetric dissimilarity measure. The ability to use an arbitrary dissimilarity measure with the $k$-medoids algorithm addresses the first shortcoming of $k$-means discussed above. Moreover, unlike $k$-means, the cluster centers in $k$-medoids (i.e. the medoids) must be points in the dataset, thus addressing the interpretability problems of $k$-means clustering.

Despite its advantages, $k$-medoids clustering is less popular than $k$-means due to its computational cost. Problem 1 is NP-hard in general [43], although heuristic solutions exist. Current state-of-the-art heuristic $k$-medoids algorithms scale quadratically in the dataset size in each iteration. However, they are still significantly slower than $k$-means, which scales linearly in dataset size in each iteration.

Partitioning Around Medoids (PAM) [20, 21] is one of the most widely used heuristic algorithms for $k$-medoids clustering, largely because it produces the best clustering quality [42, 43]. PAM is split into two subroutines: BUILD and SWAP. First, in the BUILD step, PAM aims to find an initial set of $k$ medoids by greedily and iteratively selecting points that minimize the $k$-medoids clustering loss (Equation (1)). Next, in the SWAP step, PAM considers all $k(n-k)$ possible pairs of medoid and non-medoid points and swaps the pair that reduces the loss the most. The SWAP step is repeated until no further improvements can be made by swapping medoids with non-medoids. As noted above, PAM has been empirically shown to produce better results than other popular $k$-medoids clustering algorithms. However, the BUILD step and each of the SWAP steps require $O(kn^2)$ distance evaluations and can be prohibitively expensive to run, especially for large datasets or when the distance evaluations are themselves expensive (e.g. for edit distance between long strings).

Randomized algorithms like CLARA [21] and CLARANS [37] have been proposed to improve computational efficiency, but result in worse clustering quality. More recently, [43] proposed a deterministic algorithm, dubbed FastPAM1, that guarantees the same output as PAM but improves the complexity to $O(n^2)$. However, the factor $O(k)$ improvement becomes less important when the sample size $n$ is large and the number of medoids $k$ is small compared to $n$. Throughout the rest of this work, we treat $k$ fixed and assume $k \ll n$.

**Additional related work:** Many other $k$-medoids algorithms exist. These algorithms can generally be divided into those that agree with or produce comparable results to PAM (matching state-of-the-art clustering quality, such as FastPAM and FastPAM1 [44]) and other randomized algorithms that sacrifice clustering quality for runtime (such as CLARA and CLARANS). [41] proposed a $k$-means-like algorithm that alternates between reassigning the points to their closest medoid and recomputing the medoid for each cluster until the $k$-medoids clustering loss can no longer be improved. Other

proposals include optimizations for Euclidean space and tabu search heuristics [12]. Recent work has also focused on distributed PAM, where the dataset cannot fit on one machine [45]. All of these algorithms, however, scale quadratically in dataset size or concede the final clustering quality for improvements in runtime. In an alternate approach for the single medoid problem, trimed [36], scales sub-quadratically in dataset size but exponentially in the dimensionality of the points. Other recent work [2] attempts to minimize the number of *unique* pairwise distances. Similarly, [27, 4] attempt to adaptively estimate these distances or coordinate-wise distances in specific settings.

**Contributions:** In this work, we propose a novel randomized $k$-medoids algorithm, called Bandit-PAM, that runs significantly faster than state-of-the-art $k$-medoids algorithms and achieves the same clustering results with high probability. Modeled after PAM, BanditPAM reduces the complexity on the sample size $n$ from $O(n^2)$ to $O(n \log n)$, for the BUILD step and each SWAP step, under reasonable assumptions that hold in many practical datasets. We empirically validate our results on several large, real-world datasets and observe that BanditPAM provides a reduction of distance evaluations of up to 200x while returning the same results as PAM and FastPAM1. We also release a high-performance C++ implementation of BanditPAM, callable from Python, which runs 4x faster than the state-of-the-art FastPAM1 implementation on the full MNIST dataset ($n = 70,000$) – without precomputing and caching the $O(n^2)$ pairwise distances as in FastPAM1.

Intuitively, BanditPAM works by recasting each step of PAM from a *deterministic computational problem* to a *statistical estimation problem*. In the BUILD step assignment of the $l$th medoid, for example, we need to choose the point amongst all $n - l$ non-medoids that will lead to the lowest overall loss (Equation (1)) if chosen as the next medoid. Thus, we wish to find $x$ that minimizes

$$L(\mathcal{M}; x) = \sum_{j=1}^{n} \min_{m \in \mathcal{M} \cup \{x\}} d(m, x_j) =: \sum_{j=1}^{n} g(x_j), \qquad (2)$$

where $g(\cdot)$ is a function that depends on $\mathcal{M}$ and $x$. Eq. (2) shows that the loss of a new medoid assignment $L(\mathcal{M}; x)$ can be written as the summation of the value of the function $g(\cdot)$ evaluated on all $n$ points in the dataset. Though approaches such as PAM and FastPAM1 compute $L(\mathcal{M}; x)$ exactly for each $x$, BanditPAM *adaptively estimates* this quantity by sampling reference points $x_j$ for the most promising candidates. Indeed, computing $L(\mathcal{M}; x)$ exactly for every $x$ is not required; promising candidates can be estimated with higher accuracy (by computing $g$ on more reference points $x_j$) and less promising ones can be discarded early without expending further computation.

To design the adaptive sampling strategy, we show that the BUILD step and each SWAP iteration can be formulated as a best-arm identification problem from the multi-armed bandits (MAB) literature [3, 13, 17, 18]. In the typical version of the best-arm identification problem, we have $m$ arms. At each time step $t = 0, 1, ...$, we decide to pull an arm $A_t \in \{1, \cdots, m\}$, and receive a reward $R_t$ with $E[R_t] = \mu_{A_t}$. The goal is to identify the arm with the largest expected reward with high probability with the fewest number of total arm pulls. In the BUILD step of BanditPAM, we view each candidate medoid $x$ as an arm in a best-arm identification problem. The arm parameter corresponds to $\frac{1}{n} \sum_j g(x_j)$ and pulling an arm corresponds to computing the loss $g$ on a randomly sampled data point $x_j$. Using this reduction, the best candidate medoid can be estimated using existing best-arm algorithms like the Upper Confidence Bound (UCB) algorithm [25] and successive elimination [14].

The idea of algorithm acceleration by converting a computational problem into a statistical estimation problem and designing the adaptive sampling procedure via multi-armed bandits has witnessed some recent success [10, 22, 29, 19, 6, 48]. In the context of $k$-medoids clustering, previous work [5, 7] has considered finding the *single* medoid of a set points (i.e. the 1-medoid problem). In these works, the 1-medoid problem was also formulated as a best-arm identification problem, with each point corresponding to an arm and its average distance to other points corresponding to the arm parameter.

While the 1-medoid problem considered in prior work can be solved exactly, the $k$-medoids problem is NP-Hard and is therefore only tractable with heuristic solutions. Hence, this paper focuses on improving the computational efficiency of an existing heuristic solution, PAM, that has been empirically observed to be superior to other techniques. Moreover, instead of having a single best-arm identification problem as in the 1-medoid problem, we reformulate PAM as a *sequence* of best-arm problems. Our reformulation treats different objects as arms in different steps of PAM; in the BUILD step, each point corresponds to an arm, whereas in the SWAP step, each medoid-and-non-medoid pair corresponds to an arm. We notice that the intrinsic difficulties of this sequence of best-arm problems

are different from the single best-arm identification problem, which can be exploited to further speed up the algorithm. We discuss these further optimizations in Sections 5 and 6 and Appendix 1.2.

## 2 Preliminaries

For $n$ data points $\mathcal{X} = \{x_1, x_2, \cdots, x_n\}$ and a given distance function $d(\cdot, \cdot)$, the $k$-medoids problem aims to find a set of $k$ medoids $\mathcal{M} = \{m_1, \cdots, m_k\} \subset \mathcal{X}$ to minimize the overall distance of points from their closest medoids:

$$L(\mathcal{M}) = \sum_{i=1}^{n} \min_{m \in \mathcal{M}} d(m, x_i) \tag{3}$$

Note that $d$ need not satisfy symmetry, triangle inequality, or positivity. For the rest of the paper, we use $[n]$ to denote the set $\{1, \cdots, n\}$ and $|\mathcal{S}|$ to represent the cardinality of a set $\mathcal{S}$. For two scalars $a, b$, we let $a \wedge b = \min(a, b)$ and $a \vee b = \max(a, b)$.

### 2.1 Partitioning Around Medoids (PAM)

The original PAM algorithm [20, 21] first initializes the set of $k$ medoids via the BUILD step and then repeatedly performs the SWAP step to improve the loss (3) until convergence.

**BUILD:** PAM initializes a set of $k$ medoids by greedily assigning medoids one-by-one so as to minimize the overall loss (3). The first point added in this manner is the medoid of all $n$ points. Given the current set of $l$ medoids $\mathcal{M}_l = \{m_1, \cdots, m_l\}$, the next point to add $m^*$ is

$$\text{BUILD:} \quad m^* = \underset{x \in \mathcal{X} \setminus \mathcal{M}_l}{\arg\min} \frac{1}{n} \sum_{j=1}^{n} \left[ d(x, x_j) \wedge \min_{m' \in \mathcal{M}_l} d(m', x_j) \right] \tag{4}$$

**SWAP:** PAM then swaps the medoid-nonmedoid pair that would reduce the loss (3) the most among all possible $k(n-k)$ such pairs. Let $\mathcal{M}$ be the current set of $k$ medoids. Then the best medoid-nonmedoid pair $(m^*, x^*)$ to swap is

$$\text{SWAP:} \quad (m^*, x^*) = \underset{(m,x) \in \mathcal{M} \times (\mathcal{X} \setminus \mathcal{M})}{\arg\min} \frac{1}{n} \sum_{j=1}^{n} \left[ d(x, x_j) \wedge \min_{m' \in \mathcal{M} \setminus \{m\}} d(m', x_j) \right] \tag{5}$$

The second terms in (4) and (5), namely $\min_{m' \in \mathcal{M}_l} d(m', x_j)$ and $\min_{m' \in \mathcal{M} \setminus \{m\}} d(m', x_j)$, can be determined by caching the smallest and the second smallest distances from each point to the previous set of medoids, namely $\mathcal{M}_l$ in (4) and $\mathcal{M}$ in (5). Therefore, in (4) and (5), we only need to compute the distance once for each summand. As a result, PAM needs $O(kn^2)$ distance computations for the $k$ greedy searches in the BUILD step and $O(kn^2)$ distance computations for each SWAP iteration.

## 3 BanditPAM

At the core of the PAM algorithm is the $O(n^2)$ BUILD search (4), which is repeated $k$ times for initialization, and the $O(kn^2)$ SWAP search (5), which is repeated until convergence. We first show that both searches share a similar mathematical structure, and then show that such a structure can be optimized efficiently using a bandit-based randomized algorithm, thus giving rise to BanditPAM. Rewriting the BUILD search (4) and the SWAP search (5) in terms of the change in total loss yields

$$\text{BUILD:} \quad \underset{x \in \mathcal{X} \setminus \mathcal{M}_l}{\arg\min} \frac{1}{n} \sum_{j=1}^{n} \left[ \left( d(x, x_j) - \min_{m' \in \mathcal{M}_l} d(m', x_j) \right) \wedge 0 \right] \tag{6}$$

$$\text{SWAP:} \quad \underset{(m,x) \in \mathcal{M} \times (\mathcal{X} \setminus \mathcal{M})}{\arg\min} \frac{1}{n} \sum_{j=1}^{n} \left[ \left( d(x, x_j) - \min_{m' \in \mathcal{M} \setminus \{m\}} d(m', x_j) \right) \wedge 0 \right] \tag{7}$$

One may notice that the above two problems share the following similarities. First, both are searching over a finite set of parameters: $n - l$ points in the BUILD search and $k(n-k)$ swaps in the SWAP

search. Second, both objective functions have the form of an average of an $O(1)$ function evaluated over a finite set of reference points. We formally describe the shared structure:

$$\text{Shared Problem:} \quad \underset{x \in \mathcal{S}_{\text{tar}}}{\arg\min} \frac{1}{|\mathcal{S}_{\text{ref}}|} \sum_{x_j \in \mathcal{S}_{\text{ref}}} g_x(x_j) \tag{8}$$

for target points $\mathcal{S}_{\text{tar}}$, reference points $\mathcal{S}_{\text{ref}}$, and an objective function $g_x(\cdot)$ that depends on the target point $x$. Then both BUILD and SWAP searches can be written as instances of Problem (8) with:

$$\text{BUILD:} \ \ \mathcal{S}_{\text{tar}} = \mathcal{X} \setminus \mathcal{M}_l, \ \ \mathcal{S}_{\text{ref}} = \mathcal{X}, \ \ g_x(x_j) = \left( d(x, x_j) - \min_{m' \in \mathcal{M}_l} d(m', x_j) \right) \wedge 0, \tag{9}$$

$$\text{SWAP:} \ \ \mathcal{S}_{\text{tar}} = \mathcal{M} \times (\mathcal{X} \setminus \mathcal{M}), \ \ \mathcal{S}_{\text{ref}} = \mathcal{X}, \ \ g_x(x_j) = \left( d(x, x_j) - \min_{m' \in \mathcal{M} \setminus \{m\}} d(m', x_j) \right) \wedge 0. \tag{10}$$

Crucially, in the SWAP search, each *pair* of medoid-and-non-medoid points $(m, x)$ is treated as one target point in $\mathcal{S}_{\text{tar}}$ in our formulation.

## 3.1 Adaptive search for the shared problem

Recall that the computation of $g(x_j)$ is $O(1)$. A naive, explicit method would require $O(|\mathcal{S}_{\text{tar}}||\mathcal{S}_{\text{ref}}|)$ computations of $g(x_j)$ to solve Problem (8). However, as shown in previous works [5, 6], a randomized search would return the correct result with high confidence in $O(|\mathcal{S}_{\text{tar}}| \log |\mathcal{S}_{\text{ref}}|)$ computations of $g(x_j)$. Specifically, for each target $x$ in Problem (8), let $\mu_x = \frac{1}{|\mathcal{S}_{\text{ref}}|} \sum_{x_j \in \mathcal{S}_{\text{ref}}} g_x(x_j)$ denote its objective function. Computing $\mu_x$ exactly takes $O(|\mathcal{S}_{\text{ref}}|)$ computations of $g(x_j)$, but we can instead estimate $\mu_x$ with fewer computations by drawing $J_1, J_2, ..., J_{n'}$ independent samples uniformly with replacement from $[|\mathcal{S}_{\text{ref}}|]$. Then, $E[g(x_{J_i})] = \mu_x$ and $\mu_x$ can be estimated as $\hat{\mu}_x = \frac{1}{n'} \sum_{i=1}^{n'} g(x_{J_i})$, where $n'$ determines the estimation accuracy. To estimate the solution to Problem (8) with high confidence, we can then choose to sample different targets in $\mathcal{S}_{\text{tar}}$ to different degrees of accuracy. Intuitively, promising targets with small values of $\mu_x$ should be estimated with high accuracy, while less promising ones can be discarded without being evaluated on too many reference points.

The specific adaptive estimation procedure is described in Algorithm 1. It can be viewed as a batched version of the conventional UCB algorithm [25, 48] combined with successive elimination [14], and is straightforward to implement. Algorithm 1 uses the set $\mathcal{S}_{\text{solution}}$ to track all potential solutions to Problem (8); $\mathcal{S}_{\text{solution}}$ is initialized as the set of all target points $\mathcal{S}_{\text{tar}}$. We will assume that, for a fixed target point $x$ and a randomly sampled reference point $x_J$, the random variable $Y = g_x(x_J)$ is $\sigma_x$-sub-Gaussian for some known parameter $\sigma_x$. Then, for each potential solution $x \in \mathcal{S}_{\text{solution}}$, Algorithm 1 maintains its mean objective estimate $\hat{\mu}_x$ and confidence interval $C_x$, where $C_x$ depends on the exclusion probability $\delta$ as well as the parameter $\sigma_x$. We discuss the sub-Gaussianity parameters and possible relaxations of this assumption in Sections 4 and 6 and Appendix 2.1.

In each iteration, a new batch of reference points $\mathcal{S}_{\text{ref\_batch}}$ is evaluated for all potential solutions in $\mathcal{S}_{\text{solution}}$, making the estimate $\hat{\mu}_x$ more accurate. Based on the current estimate, if a target's lower confidence bound $\hat{\mu}_x - C_x$ is greater than the upper confidence bound of the most promising target $\min_y (\hat{\mu}_y + C_y)$, we remove it from $\mathcal{S}_{\text{solution}}$. This process continues until there is only one point in $\mathcal{S}_{\text{solution}}$ or until we have sampled more reference points than in the whole reference set. In the latter case, we know that the difference between the remaining targets in $\mathcal{S}_{\text{solution}}$ is so subtle that an exact computation is more efficient. We then compute those targets' objectives exactly and return the best target in the set.

## 3.2 Algorithmic details

**Estimation of each $\sigma_x$:** BanditPAM uses Algorithm 1 in both the BUILD step and each SWAP iteration, with input parameters specified in (9) and (10). In practice, $\sigma_x$ is not known *a priori* and we estimate $\sigma_x$ for each $x \in |\mathcal{S}_{\text{tar}}|$ from the data. In the first batch of sampled reference points in Algorithm 1, we estimate each $\sigma_x$ as:

$$\sigma_x = \text{STD}_{y \in \mathcal{S}_{\text{ref\_batch}}} g_x(y) \tag{11}$$

where STD denotes standard deviation. Intuitively, this allows for smaller confidence intervals in later iterations, especially in the BUILD step, when the average arm returns to become smaller as we

**Algorithm 1** Adaptive-Search ( $\mathcal{S}_{\text{tar}}, \mathcal{S}_{\text{ref}}, g_x(\cdot), B, \delta, \sigma_x$ )

---
1: $\mathcal{S}_{\text{solution}} \leftarrow \mathcal{S}_{\text{tar}}$                ▷ Set of potential solutions to Problem (8)
2: $n_{\text{used\_ref}} \leftarrow 0$                        ▷ Number of reference points evaluated
3: For all $x \in \mathcal{S}_{\text{tar}}$, set $\hat{\mu}_x \leftarrow 0, C_x \leftarrow \infty$     ▷ Initial mean and confidence interval for each arm
4: **while** $n_{\text{used\_ref}} < |\mathcal{S}_{\text{ref}}|$ and $|\mathcal{S}_{\text{solution}}| > 1$ **do**
5:      Draw a batch samples of size $B$ with replacement from reference $\mathcal{S}_{\text{ref\_batch}} \subset \mathcal{S}_{\text{ref}}$
6:      **for all** $x \in \mathcal{S}_{\text{solution}}$ **do**
7:          $\hat{\mu}_x \leftarrow \frac{n_{\text{used\_ref}}\hat{\mu}_x + \sum_{y \in \mathcal{S}_{\text{ref\_batch}}} g_x(y)}{n_{\text{used\_ref}} + B}$              ▷ Update running mean
8:          $C_x \leftarrow \sigma_x \sqrt{\frac{\log(\frac{1}{\delta})}{n_{\text{used\_ref}} + B}}$             ▷ Update confidence interval
9:      $\mathcal{S}_{\text{solution}} \leftarrow \{x : \hat{\mu}_x - C_x \leq \min_y(\hat{\mu}_y + C_y)\}$ ▷ Remove points that can no longer be solution
10:     $n_{\text{used\_ref}} \leftarrow n_{\text{used\_ref}} + B$
11: **if** $|\mathcal{S}_{\text{solution}}| = 1$ **then**
12:      **return** $x^* \in \mathcal{S}_{\text{solution}}$
13: **else**
14:      Compute $\mu_x$ exactly for all $x \in \mathcal{S}_{\text{solution}}$
15:      **return** $x^* = \arg\min_{x \in \mathcal{S}_{\text{solution}}} \mu_x$

---

add more medoids (since we are taking the minimum over a larger set on the RHS of Eq. (4)). We also allow for arm-dependent $\sigma_x$, as opposed to a fixed global $\sigma$, which allows for narrower confidence intervals for arms whose returns are heavily concentrated (e.g. distant outliers). Empirically, this results in significant speedups and results in fewer arms being computed exactly (Line 14 in Algorithm 1). In all experiments, the batch size $B$ is set to 100 and the error probability $\delta$ is set to $\delta = \frac{1}{1000|\mathcal{S}_{\text{tar}}|}$. Empirically, these values of batch size and this setting of $\delta$ are such that BanditPAM recovers the same results in PAM in almost all cases.

**Combination with FastPAM1:** We also combine BanditPAM with the FastPAM1 optimization [43]. We discuss this optimization in Appendix 1.2.

## 4 Analysis of the Algorithm

The goal of BanditPAM is to track the optimization trajectory of the standard PAM algorithm, ultimately identifying the same set of $k$ medoids with high probability. In this section, we formalize this statement and provide bounds on the number of distance computations required by BanditPAM. We begin by considering a single call to Algorithm 1 and showing it returns the correct result with high probability. We then repeatedly apply Algorithm 1 to track PAM's optimization trajectory throughout the BUILD and SWAP steps.

Consider a single call to Algorithm 1 and suppose $x^* = \arg\min_{x \in \mathcal{S}_{\text{tar}}} \mu_x$ is the optimal target point. For another target point $x \in \mathcal{S}_{\text{tar}}$, let $\Delta_x := \mu_x - \mu_{x^*}$. To state the following results, we will assume that, for a fixed target point $x$ and a randomly sampled reference point $x_J$, the random variable $Y = g_x(x_J)$ is $\sigma_x$-sub-Gaussian for some known parameter $\sigma_x$. In practice, one can estimate each $\sigma_x$ by performing a small number of distance computations as described in Section 3.2. Allowing $\sigma_x$ to be estimated separately for each arm is beneficial in practice, as discussed in Section 6. With these assumptions, the following theorem is proved in Appendix 3:

**Theorem 1.** *For $\delta = n^{-3}$, with probability at least $1 - \frac{2}{n}$, Algorithm 1 returns the correct solution to* (6) *(for a BUILD step) or* (7) *(for a SWAP step), using a total of $M$ distance computations, where*

$$E[M] \leq 4n + \sum_{x \in \mathcal{X}} \min \left[ \frac{12}{\Delta_x^2} (\sigma_x + \sigma_{x^*})^2 \log n + B, 2n \right].$$

Intuitively, Theorem 1 states that with high probability, each step of BanditPAM returns the same result as PAM. For the general result, we assume that the data is generated in a way such that the mean rewards $\mu_x$ follow a sub-Gaussian distribution (see Section 6 for a discussion). Additionally, we assume that both PAM and BanditPAM place a hard constraint $T$ on the maximum number of SWAP iterations that are allowed. Informally, as long as BanditPAM finds the correct solution to the search

problem (6) at each BUILD step and to the search problem (7) at each SWAP step, it will reproduce the sequence of BUILD and SWAP steps of PAM identically and return the same set of final medoids. We formalize this statement with Theorem 2, and discuss the proof in Appendix 3. When the number of desired medoids $k$ is a constant and the number of allowed SWAP steps is small (which is often sufficient in practice, as discussed in Section 6), Theorem 2 implies that only $O(n \log n)$ distance computations are necessary to reproduce the results of PAM with high probability.

**Theorem 2.** *If BanditPAM is run on a dataset $\mathcal{X}$ with $\delta = n^{-3}$, then it returns the same set of $k$ medoids as PAM with probability $1 - o(1)$. Furthermore, the total number of distance computations $M_{\text{total}}$ required satisfies*

$$E[M_{\text{total}}] = O\left(n \log n\right).$$

**Remark 1:** While the limit on the maximum number of swap steps, $T$, may seem restrictive, it is not uncommon to place a maximum number of iterations on iterative algorithms. Furthermore, $T$ has been observed empirically to be $O(k)$ [43], consistent with our experiments in Section 5.

**Remark 2:** We note that $\delta$ is a hyperparameter governing the error rate. It is possible to prove results analogous to Theorems 1 and 2 for arbitrary $\delta$; we discuss this in Appendix 3.

**Remark 3:** Throughout this work, we have assumed that evaluating the distance between two points is $O(1)$ rather than $O(d)$, where $d$ is the dimensionality of the datapoints. If we were to include this dependence explicitly, we would have $E[M_{\text{total}}] = O(dn \log n)$ in Theorem 2. We discuss improving the scaling with $d$ in Appendix 2.4, and the explicit dependence on $k$ in Appendix 2.5.

## 5 Empirical Results

**Setup:** As discussed in Section 1, PAM has been empirically observed to produce the best results for the $k$-medoids problem in terms of clustering quality. Other existing algorithms can generally be divided into several classes: those that agree exactly with PAM (e.g. FastPAM1), those that do not agree exactly with PAM but provide comparable results (e.g. FastPAM) and other randomized algorithms that sacrifice clustering quality for runtime. In Subsection 5.1, we show that BanditPAM returns the same results as PAM, thus matching the state-of-the-art in clustering quality, and also results in better or comparable final loss when compared to other popular $k$-medoids clustering algorithms, including FastPAM [43], CLARANS [37], and Voronoi Iteration [41]. In Subsection 5.2, we demonstrate that BanditPAM scales almost linearly in the number of samples $n$ for all datasets and all metrics considered, which is superior to the quadratic scaling of PAM, FastPAM1, and FastPAM. Combining these observations, we conclude that BanditPAM matches state-of-the-art algorithms in clustering quality, while reaching its solutions much faster. In the experiments, each parameter setting was repeated 10 times with data subsampled from the original dataset and 95% confidence intervals are provided.

**Datasets:** We run experiments on three real-world datasets to validate the behavior of BanditPAM, all of which are publicly available. The MNIST dataset [26] consists of 70,000 black-and-white images of handwritten digits, where each digit is represented as a 784-dimensional vector. On MNIST, We consider two distance metrics: $l_2$ distance and cosine distance. The scRNA-seq dataset contains the gene expression levels of 10,170 different genes in each of 40,000 cells after standard filtering. On scRNA-seq, we consider $l_1$ distance, which is recommended [38]. The HOC4 dataset from Code.org [11] consists of 3,360 unique solutions to a block-based programming exercise. Solutions to the programming exercise are represented as abstract syntax trees (ASTs), and we consider the tree edit distance [47] to quantify similarity between solutions.

### 5.1 Clustering/loss quality

Figure 1 (a) shows the relative losses of algorithms with respect to the loss of PAM. BanditPAM and three other baselines: FastPAM [43], CLARANS [37], and Voronoi Iteration [41]. We clarify the distinction between FastPAM and FastPAM1: both are $O(n^2)$ in each SWAP step but FastPAM1 is guaranteed to return the same solution as PAM while FastPAM is not. In these experiments, BanditPAM returns the same solution as PAM and hence has loss ratio 1. FastPAM has a comparable performance, while the other two algorithms are significantly worse.

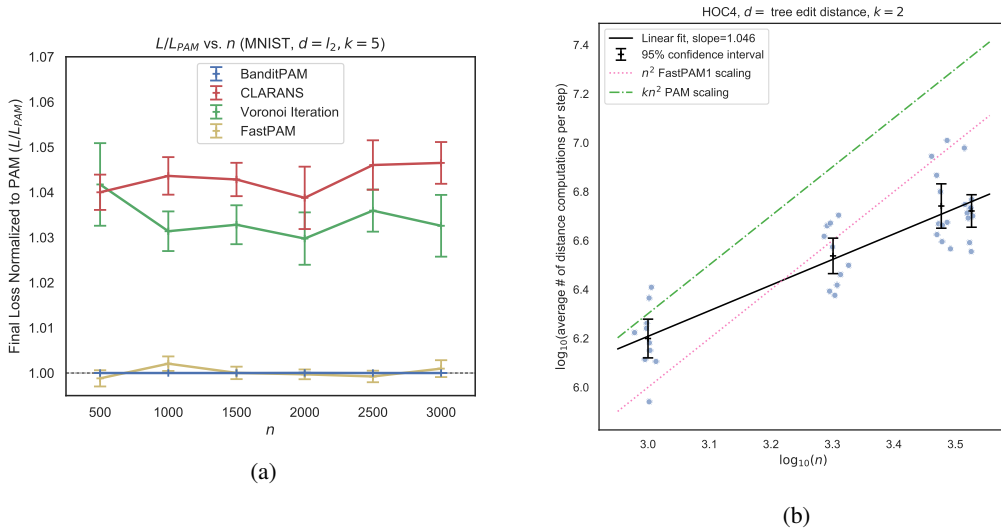

(a)

(b)

Figure 1: (a) Clustering loss relative to PAM loss. Data is subsampled from MNIST, sample size $n$ varies from $500$ to $3,000$, $k = 5$, and 95% confidence intervals are provided. BanditPAM always returns the same solution as PAM and hence has loss ratio exactly 1, as does FastPAM1 (omitted for clarity). FastPAM also demonstrates comparable final loss, while the other two algorithms are significantly worse. (b) Average number of distance evaluations per iteration vs. sample size $n$ for HOC4 and tree edit distance, with $k = 2$, on a log-log plot. Reference lines for PAM and FastPAM1 are also shown. BanditPAM scales better than PAM and FastPAM1 and is significantly faster for large datasets.

## 5.2 Scaling with $n$ for different datasets, distance metric, and $k$ values

We next consider the runtime per iteration of BanditPAM, especially in comparison to PAM and FastPAM1. To calculate the runtime per iteration of BanditPAM, we divide the total wall clock time by the number of SWAP iterations plus 1, where each SWAP step has expected complexity $O(kn \log n)$ and the plus 1 accounts for the $O(kn \log n)$ complexity of all $k$ BUILD steps. Figure 2 demonstrates the runtime per iteration of BanditPAM versus $n$ on a log-log plot. The slopes for the lines of best fit for (a) $k = 5$ and (b) $k = 10$ are 0.984 and 0.922, respectively, indicating the scaling is linear in $n$ for different values of $k$.

Figure 3 demonstrates the runtime per iteration of BanditPAM for other datasets and metrics. On MNIST with cosine distance (a), the slope of the line of best fit is 1.007. On the scRNA-seq dataset with $l_1$ distance (b), the slope of the line of best fit is 1.011. These results validate our theory that BanditPAM takes an almost linear number of distance evaluations per iteration for different datasets and metrics.

Because the exact runtime of BanditPAM and other baselines depends on implementation details such as programming language, we also analyze the number of distance evaluations required by each algorithm. Indeed, a profile of BanditPAM program reveals that it spends over 98% of its wall clock time computing distances; as such, the number of distance evaluations provides a reasonable proxy for complexity of BanditPAM. For the other baselines PAM and FastPAM1, the number of distance evaluations is expected to be exactly $kn^2$ and $n^2$, respectively, in each iteration. Figure 1 (b) demonstrates the number of distance evaluations per iteration of BanditPAM with respect to $n$. The slope of the line of best fit on the log-log plot is 1.046, which again indicates that BanditPAM scales linearly in dataset size even for more exotic objects and metrics, such as trees and tree edit distance.

## 6 Discussion and Conclusions

In this work, we proposed BanditPAM, a randomized algorithm for the $k$-medoids problem that matches state-of-the-art approaches in clustering quality while achieving a reduction in complexity

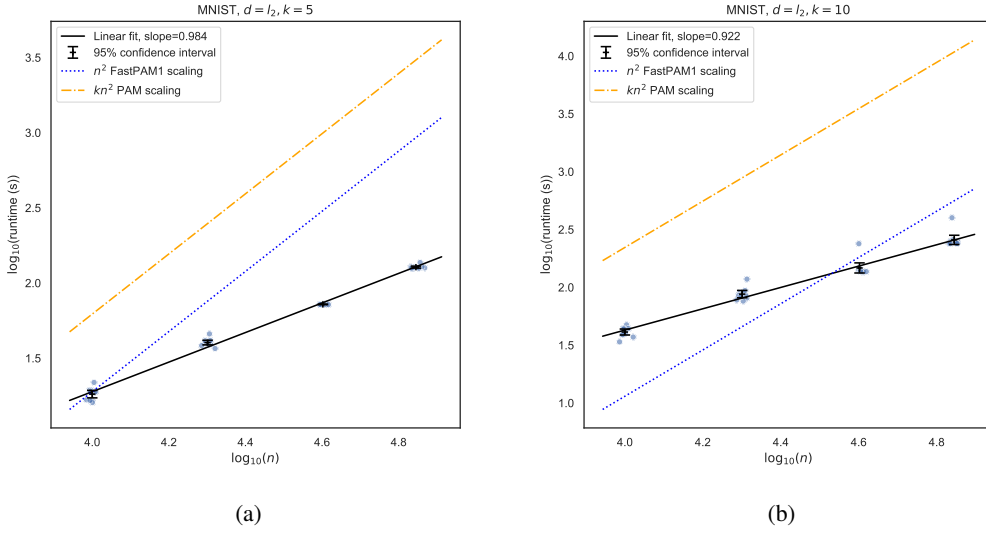

(a)                                                    (b)

Figure 2: Average runtime per iteration vs. sample size $n$ for MNIST and $l_2$ distance with (a) $k = 5$ and (b) $k = 10$, on a log-log scale. Lines of best fit (black) are plotted, as are reference lines demonstrating the expected scaling of PAM and FastPAM1.

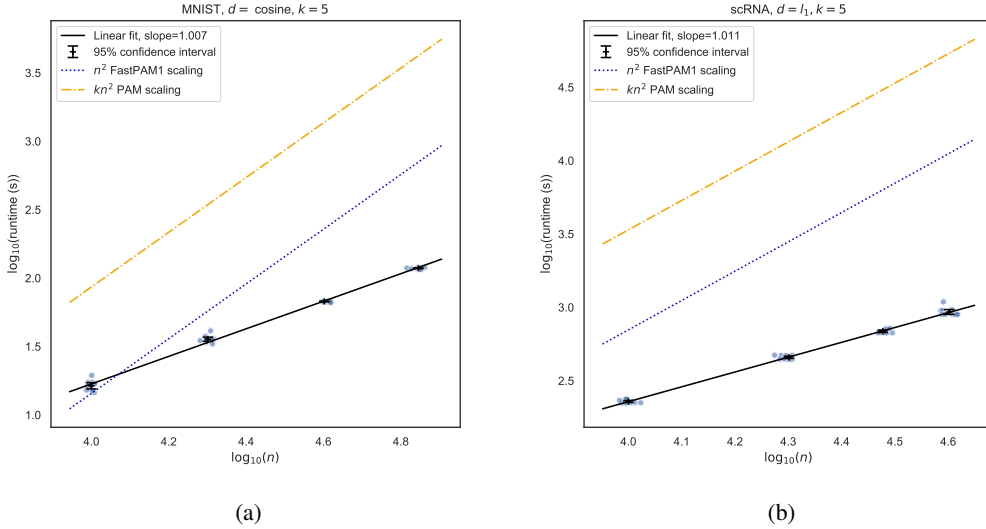

(a)                                                    (b)

Figure 3: Average runtime per iteration vs. sample size $n$, for (a) MNIST and cosine distance and (b) scRNA-seq and $l_1$ distance, with $k = 5$. Lines of best fit (black) are plotted, as are reference lines demonstrating the expected scaling of PAM and FastPAM1.

from $O(n^2)$ to $O(n \log n)$ under certain assumptions. In our experiments, the randomly sampled distances have an empirical distribution similar to a Gaussian (Appendix Figures 3-4), justifying the sub-Gaussian assumption in Section 4. We also observe that the the sub-Gaussian parameters are different across steps and target points (Appendix Figures 1), justifying the adaptive estimation of the sub-Gaussianity parameters in Subsection 3.2. Additionally, the empirical distribution of the true arm parameters (Appendix Figure 2) appears to justify the distributional assumption of $\mu_x$s in Section 4.

## Broader Impact

BanditPAM accelerates finding solutions to the $k$-medoids problem while producing comparable – and usually equivalent – final cluster assignments. Our work enables the discovery of high-quality medoid assignments in very large datasets, including some on which prior algorithms were prohibitively expensive. A potential negative consequence of this is that practitioners may be incentivized to gather and store larger amounts of data now that it can be meaningfully processed, in a phenomenon more generally described as induced demand [15]. This incentive realignment could potentially result in negative externalities such as an increase in energy consumption and carbon footprints.

Our application to the HOC4 dataset suggests a method for scaling personalized feedback to individual students in online courses. If limited resources are available, instructors can choose to provide feedback on just the *medoids* of submitted solutions instead of exhaustively providing feedback on *every* unique solution, of which there may be several thousand. Instructors can then refer individual students to the feedback provided for their closest medoid. We anticipate that this approach can be applied generally for students of Massive Open Online Courses (MOOCs), thereby enabling more equitable access to education and personalized feedback for students.

We also anticipate, however, that BanditPAM will enable several beneficial applications in other fields such as biomedicine and and fairness. For example, the evolutionary pathways of infectious diseases could possibly be constructed from the medoids of genetic sequences available at a given point in time, if prior temporal information about these sequences' histories is not available. Similarly, the medoids of patients infected in a disease outbreak may shed ligh the origins of outbreaks, as did prior analyses of cholera outbreaks using Voronoi Iteration [9]. As discussed in Section 6, our application to the HOC4 data also demonstrates the utility of BanditPAM in online education. In particular, especially with recent interest in online learning, we hope that our work will improve the quality of online learning for students worldwide.

## Acknowledgments and Disclosure of Funding

This research was funded in part by JPMorgan Chase & Co. Any views or opinions expressed herein are solely those of the authors listed, and may differ from the views and opinions expressed by JPMorgan Chase & Co. or its affiliates. This material is not a product of the Research Department of J.P. Morgan Securities LLC. This material should not be construed as an individual recommendation for any particular client and is not intended as a recommendation of particular securities, financial instruments or strategies for a particular client. This material does not constitute a solicitation or offer in any jurisdiction. M.Z. was also supported by NIH grant R01 MH115676. We would like to thank Eric Frankel for help with the C++ implementation of BanditPAM.

## Footnotes

[1]https://github.com/ThrunGroup/BanditPAM

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
