[Supplementary Material]

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

# Appendix

## 1 Additional Discussions

### 1.1 FastPAM1 optimization

Algorithm 1 can also be combined with the FastPAM1 optimization from [43] to reduce the computation in each SWAP iteration. For a given candidate swap $(m, x)$, we rewrite $g_{(m,x)}(x_j)$ from Equation (10) as:

$$g_{m,x}(x_j) = -d_1(x_j) + \mathbb{1}_{x_j \notin \mathcal{C}_m} \min[d_1(x_j), d(x, x_j)] + \mathbb{1}_{x_j \in \mathcal{C}_m} \min[d_2(x_j), d(x, x_j)] \quad (12)$$

where $\mathcal{C}_m$ denotes the set of points whose closest medoid is $m$ and $d_1(x_j)$ and $d_2(x_j)$ are the distance from $x_j$ to its nearest and second nearest medoid, respectively, before the swap is performed. We cache the values $d_1(x_j), d_2(x_j)$, and the cluster assignments $\mathcal{C}_m$ so that Equation (12) no longer depends on $m$ and instead depend only on $\mathbb{1}_{\{x_j \in \mathcal{C}_m\}}$, which is cached. This allows for an $O(k)$ speedup in each SWAP iteration since we do not need to recompute Equation 12 for each of the $k$ distinct medoids (values of $m$).

### 1.2 Value of re-estimating each $\sigma_x$

Appendix Figure 1: Boxplot showing the min, max, and each quartile for the set of all $\sigma_x$ estimates for the full MNIST dataset, in the BUILD step.

The theoretical results in Section 4 and empirical results in Section 5 suggest that BanditPAM scales almost linearly in dataset size for a variety of real-world datasets and commonly used metrics. One may also ask if Lines 7-8 of Algorithm 1, in which we re-estimate each $\sigma_x$ from the data, are necessary. In some sense, we treat the set of $\{\sigma_x\}$ as adaptive in two different ways: $\sigma_x$ is calculated on a *per-arm* basis (hence the subscript $x$), as well recalculated in each BUILD and SWAP iteration. In practice, we observe that re-estimating each $\sigma_x$ for each sequential call to Algorithm 1 significantly improves the performance of our algorithm. Figure 1 describes the distribution of estimate $\sigma_x$ for the MNIST data at different stages of the BUILD step. The median $\sigma_x$ drops dramatically after the first medoid has been assigned and then steadily decreases, as indicated by the orange lines, and suggests that each $\sigma_x$ should be recalculated at every assignment step. Furthermore, the whiskers demonstrate significant variation amongst the $\sigma_x$ in a given assignment step and suggest that having arm-dependent $\sigma_x$ parameters is necessary. Without these modifications to our algorithm, we find that the confidence intervals used by BanditPAM (Line 8) are unnecessarily large and cause computation to be expended needlessly as it becomes harder to identify the best target points. Intuitively, this is due to the much larger confidence intervals that make it harder to distinguish between arms' mean returns. For a more detailed discussion of the distribution of $\sigma_x$ and examples where the assumptions of Theorem 2 are violated, we refer the reader to Appendix 1.3.

## 1.3 Violation of distributional assumptions

In this section, we investigate the robustness of BanditPAM to violations of the assumptions in Theorem 2 on an example dataset and provide intuitive insights into the degradation of scaling. We create a new dataset from the scRNA dataset by projecting each point onto the top 10 principal components of the dataset; we call the dataset of projected points scRNA-PCA. Such a transformation is commonly used in prior work; the most commonly used distance metric between points is then the $l_2$ distance [31].

Figure 2 shows the distribution of arm parameters for various (dataset, metric) pairs in the first BUILD step. In this step, the arm parameter corresponds to the mean distance from the point (the arm) to every other point. We note that the true arm parameters in scRNA-PCA are more heavily concentrated about the minimum than in the other datasets. Intuitively, we have projected the points from a 10,170-dimensional space into a 10-dimensional one and have lost significant information in the process. This makes many points appear "similar" in the projected space.

Figures 3 and 4 show the distribution of arm rewards for 4 arms (points) in MNIST and scRNA-PCA, respectively, in the first BUILD step. We note that the examples from scRNA-PCA display much larger tails, suggesting that their sub-Gaussianity parameters $\sigma_x$ are very high.

Together, these observations suggest that the scRNA-PCA dataset may violate the assumptions of Theorems 1 and 2 and hurt the scaling of BanditPAM with $n$, as measured by the number of distance calls per iteration as in Section 5. Figure 5 demonstrates the scaling of BanditPAM with $n$ on scRNA-PCA. The slope of the line of best fit is 1.204, suggesting that BanditPAM scales as approximately $O(n^{1.2})$ in dataset size. We note that this is higher than the exponents suggested for other datasets by Figures 2 and 3, likely to the different distributional characteristics of the arm means and their spreads.

We note that, in general, it may be possible to characterize the distribution of arm returns $\mu_i$ at and the distribution of $\sigma_x$, the sub-Gaussianity parameter, at every step of BanditPAM, from properties of the data-generating distribution, as done for several distributions in [5]. We leave this more general problem, as well as its implications for the complexity of our BanditPAM, to future work.

Appendix Figure 2: Histogram of true arm parameters, $\mu_i$, for 1000 randomly sampled arms in the first BUILD step of various datasets. For scRNA-PCA with $d = l_2$ (bottom right), the arm returns are much more sharply peaked about the mininum than for the other datasets. In plots where the bin widths are less than 1, the frequencies can be greater than 1.

Appendix Figure 3: Example distribution of rewards for 4 points in MNIST in the first BUILD step. The minimums and maximums are indicated with red markers.

Appendix Figure 4: Example distribution of rewards for 4 points in scRNA-PCA in the first BUILD step. The minimums and maximums are indicated with red markers. The distributions shown here are more heavy-tailed than in Figure 3. In plots where the bin widths are less than 1, the frequencies can be greater than 1.

Appendix Figure 5: Average number of distance calls per iteration vs. $n$, for scRNA-PCA and $l_2$ distance on a log-log scale. The line of best fit (black) are plotted, as are reference lines demonstrating the expected scaling of PAM (red).

## 2 Future Work

There are several ways in which BanditPAM could be improved or made more impactful. In this work, we chose to implement a UCB-based algorithm to find the medoids of a dataset. Other best-arm-identification approaches, however, could also be used for this problem. It may also be possible to generalize a recent single-medoid approach, Correlation-Based Sequential Halving [7], to more than 1 medoid, especially to relax the sub-Gaussianity assumptions (discussed further in Appendix 2.1). Though we do not have reason to suspect an algorithmic speedup (as measured by big-$O$), we may see constant factor or wall clock time improvements. We also note that it may be possible to prove the optimality of BanditPAM in regards to algorithmic complexity, up to constant factors, using techniques from [5] that were developed for sample-efficiency guarantees in hypothesis testing.

We also note that it may be possible to improve the theoretical bounds presented in Theorem 1; indeed, in experiments, a much larger error threshold $\delta$ was acceptable, which suggests that the bounds are weak; we discuss the hyperparameter $\delta$ further in Appendix 2.3.

### 2.1 Relaxing the sub-Gaussianity assumption

An alternate approach using bootstrap-based bandits [46, 24, 23] could be valuable in relaxing the distributional assumptions on the data that the quantities of interest are $\sigma$-sub-Gaussian. Alternatively, if a bound on the distances is known, it may be possible to avoid the estimation of $\sigma_x$ by using the empirical Bernstein inequality to bound the number of distance computations per point, similar to how Hoeffding's inequality was used in the proof of Theorem 1. It may also be possible to use a related method from [1] to avoid these statistical assumptions entirely.

### 2.2 Intelligent Cache Design

The existing implementation does not cache pairwise distance computations, despite the fact that BanditPAM spends upwards of 98% of its runtime in evaluating distances, particularly when such distances are expensive to compute. This is in stark contrast to the state-of-the-art implementations of PAM, FastPAM1, and FastPAM1, which precompute and cache the entire $n^2$ distance matrix before any medoid assignments are made.

It should be possible to implement a cache in BanditPAM that would dramatically reduce wall-clock-time requirements. Furthermore, it may be possible to cache only $O(n \log n)$ pairwise distances, instead of all $n^2$ distances. This could be done, for example, by fixing an ordering of the reference points to be used in each call to Algorithm 1. Since, on average, only $O(\log n)$ reference points are required for each target point, it should not be necessary to cache all $n^2$ pairwise distances. Furthermore, the same cache could be used across different calls to Algorithm 1, particularly since we did not require independence of the sampling of the reference points in the proof of Theorem 2. Finally, it may be possible to reduce this cache size further by using techniques from [35].

### 2.3 Approximate version of BanditPAM

We note that BanditPAM is a randomized algorithm which requires the specification of the hyperparameter $\delta$. Intuitively, $\delta$ governs the error probability that BanditPAM returns a suboptimal target point $x$ in any call to Algorithm 1. The error parameter $\delta$ suggests the possibility for an approximate version of BanditPAM that may not required to return the same results as PAM. If some concessions in final clustering loss are acceptable, $\delta$ can be increased to improve the runtime of BanditPAM. An analysis of the tradeoff between the final clustering loss and runtime of BanditPAM, governed by $\delta$, is left to future work. It may also be possible to combine the techniques in [39] with BanditPAM to develop an approximate algorithm.

### 2.4 Dependence on $d$

Throughout this work, we assumed that computing the distance between two points was an $O(1)$ operation. This obfuscates the dependence on the dimensionality of the data, $d$. If we consider computing the distance between two points an $O(d)$ computation, the complexity of BanditPAM could be expressed as $O(dn\log n)$ in the BUILD step and each SWAP iteration. Recent work [6] suggests that this could be further improved; instead of computing the difference in each of the $d$

coordinates, we may be able to adaptively sample which of the $d$ coordinates to use in our distance computations and reduce the dependence on dimensionality from $d$ to $O(\log d)$, especially in the case of sparse data.

## 2.5 Dependence on $k$

In this paper, we treated $k$ as a constant in our analysis and did not analyze the explicit dependence of BanditPAM on $k$. In experiments, we observed that the runtime of BanditPAM scaled linearly in $k$ in each call to Algorithm 1 when $k \ll n$. We also observe BanditPAM scales linearly with $k$ when $k$ is less than the number of "natural" clusters of the dataset (e.g. $\sim$10 for MNIST). However, we were able to find other parameter regimes where the scaling of Algorithm 1 with $k$ appears quadratic. Furthermore, we generally observed that the number of swap steps $T$ required for convergence was $O(k)$, consistent with [43], which could make the overall scaling of BanditPAM with $k$ superlinear when each call to Algorithm 1 is also $O(k)$. We emphasize that these are only empirical observations. We leave a formal analysis of the dependence of the overall BanditPAM algorithm on $k$ to future work.

## 3 Proofs of Theorems 1 and 2

**Theorem 1.** *For $\delta = n^{-3}$, with probability at least $1 - \frac{2}{n}$, Algorithm 1 returns the correct solution to* (6) *(for a BUILD step) or* (7) *(for a SWAP step), using a total of $M$ distance computations, where*

$$E[M] \leq 4n + \sum_{x \in \mathcal{X}} \min \left[ \frac{12}{\Delta_x^2} \left( \sigma_x + \sigma_{x^*} \right)^2 \log n + B, 2n \right].$$

*Proof.* First, we show that, with probability at least $1 - \frac{2}{n}$, all confidence intervals computed throughout the algorithm are true confidence intervals, in the sense that they contain the true parameter $\mu_x$. To see this, notice that for a fixed $x$ and a fixed iteration of the algorithm, $\hat{\mu}_x$ is the average of $n_{\text{used\_ref}}$ i.i.d. samples of a $\sigma_x$-sub-Gaussian distribution. From Hoeffding's inequality,

$$\Pr \left( |\mu_x - \hat{\mu}_x| > C_x \right) \leq 2 \exp \left( -\frac{n_{\text{used\_ref}} C_x^2}{2\sigma_x^2} \right) =: 2\delta.$$

Note that there are at most $\frac{n^2}{B} \leq n^2$ such confidence intervals computed across all target points (i.e. arms) and all steps of the algorithm, where $B$ is the batch size. If we set $\delta = 1/n^3$, we see that $\mu_x \in [\hat{\mu}_x - C_x, \hat{\mu}_x + C_x]$ for every $x$ and for every step of the algorithm with probability at least $1 - \frac{2}{n}$, by the union bound over at most $n^2$ confidence intervals.

Next, we prove the correctness of Algorithm 1. Let $x^* = \arg \min_{x \in \mathcal{S}_{\text{tar}}} \mu_x$ be the desired output of the algorithm. First, observe that the main `while` loop in the algorithm can only run $\frac{n}{B}$ times, so the algorithm must terminate. Furthermore, if all confidence intervals throughout the algorithm are correct, it is impossible for $x^*$ to be removed from the set of candidate target points. Hence, $x^*$ (or some $y \in \mathcal{S}_{\text{tar}}$ with $\mu_y = \mu_{x^*}$) must be returned upon termination with probability at least $1 - \frac{2}{n}$.

Finally, we consider the complexity of Algorithm 1. Let $n_{\text{used\_ref}}$ be the total number of arm pulls computed for each of the arms remaining in the set of candidate arms at some point in the algorithm. Notice that, for any suboptimal arm $x \neq x^*$ that has not left the set of candidate arms, we must have $C_x = \sigma_x \sqrt{\log(\frac{1}{\delta})/n_{\text{used\_ref}}}$. With $\delta = n^{-3}$ as above and $\Delta_x := \mu_x - \mu_{x^*}$, if $n_{\text{used\_ref}} > \frac{12}{\Delta_x^2} \left( \sigma_x + \sigma_{x^*} \right)^2 \log n$, then

$$2(C_x + C_{x^*}) = 2 \left( \sigma_x + \sigma_{x^*} \right) \sqrt{\log(n^3)/n_{\text{used\_ref}}} < \Delta_x = \mu_x - \mu_{x^*},$$

and

$$\begin{aligned}
\hat{\mu}_x - C_x &> \mu_x - 2C_x \\
&= \mu_{x^*} + \Delta_x - 2C_x \\
&\geq \mu_{x^*} + 2C_{x^*} \\
&> \hat{\mu}_{x^*} + C_{x^*}
\end{aligned}$$

implying that $x$ must be removed from the set of candidate arms at the end of that iteration. Hence, the number of distance computations $M_x$ required for target point $x \neq x^*$ is at most

$$M_x \leq \min \left[ \frac{12}{\Delta_x^2} \left( \sigma_x + \sigma_{x^*} \right)^2 \log n + B, 2n \right].$$

Notice that this holds simultaneously for all $x \in \mathcal{S}_{\text{tar}}$ with probability at least $1 - \frac{2}{n}$. We conclude that the total number of distance computations $M$ satisfies

$$E[M] \leq E[M | \text{ all confidence intervals are correct}] + \frac{2}{n}(2n^2)$$

$$\leq 4n + \sum_{x \in \mathcal{X}} \min \left[ \frac{12}{\Delta_x^2} \left( \sigma_x + \sigma_{x^*} \right)^2 \log n + B, 2n \right]$$

where we used the fact that the maximum number of distance computations per target point is $2n$. $\quad \square$

**Remark A1:** An analogous claim can be made for arbitrary $\delta$. For arbitrary $\delta$, the probability that all confidence intervals are true confidence intervals is at least $1 - 2n^2\delta$, and the expression for $E[M]$ becomes:

$$E[M] \leq E[M| \text{ all confidence intervals are correct}] + 4n^4\delta$$

$$\leq 4n^4\delta + \sum_{x\in\mathcal{X}} \min\left[\frac{4}{\Delta_x^2}(\sigma_x + \sigma_{x^*})^2 \log(\frac{1}{\delta}) + B, 2n\right]$$

**Theorem 2.** *If BanditPAM is run on a dataset $\mathcal{X}$ with $\delta = n^{-3}$, then it returns the same set of $k$ medoids as PAM with probability $1 - o(1)$. Furthermore, the total number of distance computations $M_{\text{total}}$ required satisfies*

$$E[M_{\text{total}}] = O(n \log n).$$

From Theorem 1, the probability that Algorithm 1 does not return the target point $x$ with the smallest value of $\mu_x$ in a single call, i.e. that the result of Algorithm 1 will differ from the corresponding step in PAM, is at most $2/n$. By the union bound over all $k + T$ calls to Algorithm 1, the probability that BanditPAM does not return the same set of $k$ medoids as PAM is at most $2(k + T)/n = o(1)$, since $k$ and $T$ are taken as constants. This proves the first claim of Theorem 2.

It remains to show that $E[M_{\text{total}}] = O(n \log n)$. Note that, if a random variable is $\sigma$-sub-Gaussian, it is also $\sigma'$-sub-Gaussian for $\sigma' > \sigma$. Hence, if we have a universal upper bound $\sigma_{\text{ub}} > \sigma_x$ for all $x$, Algorithm 1 can be run with $\sigma_{\text{ub}}$ replacing each $\sigma_x$. In that case, a direct consequence of Theorem 1 is that the total number of distance computations per call to Algorithm 1 satisfies

$$E[M] \leq 4n + \sum_{x\in\mathcal{X}} 48\frac{\sigma_{\text{ub}}^2}{\Delta_x^2}\log n + B \leq 4n + 48\left(\frac{\sigma_{\text{ub}}}{\min_x \Delta_x}\right)^2 n \log n. \tag{13}$$

Furthermore, as proven in Appendix 2 of [5], such an instance-wise bound, which depends on the $\Delta_x$s, converts to an $O(n \log n)$ bound when the $\mu_x$s follow a sub-Gaussian distribution. Moreover, since at most $k + T$ calls to Algorithm 1 are made, from (13) we see that the total number of distance computations $M_{\text{total}}$ required by BanditPAM satisfies $E[M_{\text{total}}] = O(n \log n)$.