[Reviews · NeurIPS 2020]

Review 1

Summary and Contributions: This paper addresses how to use bandits to accelerate computation for the Partitioning Around Mediods algorithm (PAM). PAM is a heuristic to solve the k-mediods problem which is in general NP-Hard. The authors follow a line of work established by previous works they’ve cited whereby, objectives that are written as a sum can be subsampled and approximated. This can lead to exponential speedups if only log(n) samples need to be taken to sufficiently approximate the sum on average. This relies on statistical assumptions on the data which the authors validate empirically. The main contribution is the Bandit-PAM algorithm, a way of accelerating the PAM algorithm from O(n^2) to O(n\log(n)) when k is O(1). Additionally, the authors provide a fast implementation of their method which may be of independent value to practitioners.

Strengths: The paper is, for the most part, clear and well written. The introduction is cleanly done and well-motivated, and it is a nice idea in general. Bandit-PAM appears to be 3.2x faster in wall clock time, compared to past work. The algorithm is decently straightforward, and the statistical assumptions made on the data are well validated in the appendix.

Weaknesses: The empirical results are confusingly presented. Leading up to them, the authors discuss how their method is faster than others. However, in experiments, they do not provide a plot or a discussion of wall-clock time speedups. They do provide a comparison of the loss achieved compared to other methods, but they achieve the same loss as FAST-PAM. This leads the reader to believe that their method does not improve over baselines, when in fact the guarantee from their analysis seems to be that they can achieve the same loss as other methods but take much less time. If this is the story they wish to tell in this work, this needs to be made more prominent. Additionally, the paper does not feel entirely self-contained. Because I am familiar with other works cited in the related works, the reduction from \sum_i 1/\Delta_i^2 to an O(n/\log(n)) speedup is immediate to me. However, for readers not familiar with Bagaria et al., 2018 this is more mysterious and ought to be covered more thoroughly, if only by example, in the appendix of this result as well. The comparison to fast-PAM1 feels like a bait and switch since a fair bit of the main body mentions it, but then it is not included in experiments and barely covered in the appendix.

Correctness: The theoretical guarantees all seem correct after a quick glance through the proof of Theorem 2, though the connection between Theorem 2 and Theorem 1 could be clearer. Additionally, in line 171, the authors state that their method is an adaptation of the standard UCB algorithm. This is not correct. In fact, their method as written is an adaptation of successive elimination instead by Even-Dar et al., 2002.

Clarity: As discussed in the weaknesses, the discussion around the results section could be much clearer. FAST-PAM1 also should be explained more carefully. Readers familiar with related work may be surprised to not see any analysis of the dimension of vectors d in this work, and the fact that the authors assume distances can be computed in O(1) time rather than O(d). Certainly, the ideas are not specific to subsampling distances, but it is worth stating this explicitly for the reader. Additionally, the dispersion parameter \simga_x is referenced before it is defined. I thought the experiments were well done and interesting. I would encourage the authors to include an additional supplementary experiment section in the appendix to include further details that cannot fit in the main body. In particular, the MOOC example seemed cool, but was not covered too thoroughly. One question in this vein I had: how were the confidence intervals computed? They seem very tight given than only 10 repetitions were performed.

Relation to Prior Work: This work follows a line of works established By Bagaria et al. using statistical tools to convert computational tasks into estimation problems. Intuitively, the idea makes a lot of sense. Often to show that a point is far away or in some way suboptimal, it is not necessary to learn a distance/an objective to high accuracy. Therefore, one can simply subsample the sum. I did not explicitly check, but I trust that this optimization does not immediately follow from previous, more general ones written, such as those in “Adaptive Monte Carlo Optimization” by Bagaria et al.

Reproducibility: Yes

Additional Feedback: - If a bound on the distances is known, the estimation of sigma_x can be avoided entirely by using a bound such as the empirical Bernstein inequality. This is likely the more statistically correct tool than first estimating the deviations and then plugging in. - Additionally, if you prefer not to make these statistical assumptions in the first place, you could use a method such as “Best of both worlds: stochastic and adversarial best-arm identification” by Abbasi-Yadkori et al which allows for adaptive speedups with fewer assumptions than those made herein. - Legends in Figures 1 and 2 are very hard to read.


Review 2

Summary and Contributions: In the k-medoids clustering problem, given a set [n] of points, the goal is to find a subset S of [n] such that the sum of the minimum distance of points in [n] to S is minimized. That is, finding S \subset [n] that minimizes \sum_{i \in [n]} min_{j \in S} d(i,j). A commonly used heuristic for finding such a set is PAM which has a greedy initialization phase (to find an initial set S) and a swap phase (to improve the set S). The initialization phase and each step of the swap phase needs to evaluate kn^2 distances --- this is improved to n^2 in other methods. The goal of this paper is to improve the number of needed distances in each step to n log(n). The proposed method is to consider a loss \mu(x) for each point x that can be added in the initialization phase and a loss \mu(x,y) each pair (x,y) that can be swaped where x \in S and y \notin S. Then the goal is to estimate the value of \mu(x) and \mu(x,y)'s and pick the best. The proposed method to solve this problem is the upper confidence bound (UCB) algorithm which needs O(n log(n)) distance evaluations to find the best point or pair in each step with a probability of 1-(2/n). In the experiments, the proposed method is compared to classical methods in terms of final loss (i.e., the clustering cost), and the number of used distances.

Strengths: The considered problem is an important one because decreasing the number of distances needed for this method can enable the usage of k-medoids in applications with large datasets.

Weaknesses: The contribution of the paper is very small. The same bandit method is used in [2] to find the medoid of a set of points with a probability of 1-o(1). This paper just argues that they can use the same method in multiple steps and use the union bound to argue that the probability of success is 1 - 2(k+T)/n, where k is the number of medoids and T is the number of swap steps. One important drawback of such union bound is that if T is large (such that 2(k+T)/n > 1), then the proposed method does not have any guarantee. Moreover, it would be good to mention the number of swap steps in the experiments and also perform the experiments with larger values of k and until convergence.

Correctness: I did not check the proofs in the supplementary material, but the claims seem correct.

Clarity: Yes, the paper is well written and clear.

Relation to Prior Work: It would be nice to discuss the connections to [2] more. For example, if the proof is similar to the proof of that paper, it should be stated.

Reproducibility: No

Additional Feedback: 1) It would be good to mention the number of swap steps in the experiments. I suspect the solution with the bandit method should deviate from the solution returned by PAM if the number of swap steps are large. It would be good to see in how many steps this happens. 2) One interesting thing to report is the actual number of distances (without redundancy) used by the algorithm. I mean we can use the distances that algorithm evaluated in previous steps in the future steps without extra cost. So if d(1,2) is evaluated in the first swap step, we can use it in the second swap step and so on. 3) Decreasing the number of distances needed by k-medoids is also considered from an active learning viewpoint. See the following paper. Aghaee, A., Ghadiri, M., & Baghshah, M. S. (2016, April). Active distance-based clustering using K-medoids. In Pacific-Asia Conference on Knowledge Discovery and Data Mining (pp. 253-264). Springer, Cham. ======= After Rebuttal Edit ======= I still think that the theoretical result is derived by easy and straightforward modifications of arguments in [2]. However, I agree that clustering is a very important topic and the proposed algorithm might be useful in practice. Considering all of these factors, my score stays unchanged.


Review 3

Summary and Contributions: The authors study k-medoids (similar to k-means, but the centers must be part of the point set, and therefore the problem can be defined on any dissimilarity measure). A popular k-medoids algorithm is partitioning around medoids (PAM), which has O(k*n^2) runtime. The authors give a bandit-based algorithm which runs in O(k*n log n) time (or O(n log n) when k is a constant), and gives the same output as PAM under reasonable assumptions about the data. The authors give theoretical arguments for the correctness of their algorithm, as well as experiments on several datasets. Specifically, PAM involves greedily finding centers which minimize the median loss, both in the BUILD (initialization) phase and the SWAP phase, which take O(n^2) time each time. Recent work shows how to use a sampling-based approach to improve the runtime of 1-medoid from O(n^2) to O(n log n) by treating the problem of finding the best medoid as a multi-arm bandit problem. The authors apply these results to k-medoid by casting both parts of the PAM algorithm into the multi-arm bandit framework. ========================= post-rebuttal ============================= I thank the authors for providing a rebuttal which addresses my concerns. I agree that I was mistaken in my review, and PAM actually is state-of-the-art if we are just talking about clustering quality. Therefore, I think the experiments section is more convincing than I originally believed. However, I agree with R1 that the experiments section should be made less confusing. For example, we can group k-medoids algorithms into two groups: those that give virtually the same output as PAM (e.g. FastPAM, FastPAM1, FastPAM2), and those that are faster, but output a solution with significantly worse quality than PAM (CLARANS, FastCLARANS, Voronoi Iteration). BanditPAM falls into the first group, so the experiments should compare to the algorithms in this group, although they can also keep Figure 1a as an aside, to show the distinction between the two groups. I agree with the authors' comments about trimed, and that it should be clarified in the paper. It will also be good to update Theorems 1 and 2 to be more general. I am updating my score from a 6 to a 7.

Strengths: The authors extend prior work on 1-medoid to k-medoids in a nontrivial way, achieving a significant reduction in runtime for the PAM algorithm. Such an improvement is a contribution to the area of k-medoid clustering research. The paper is well-written and well-organized. The broader impact is also well-done. I think the code submission is the best I’ve ever seen. The authors give detailed instruction in how to set up the requirements (including links to common errors installing packages), the easiest way to download each dataset, descriptions of each code file, and descriptions on how to reproduce every figure in their paper. The files are also commented well, and they also give another faster version of their code in C++. This is the gold standard of reproducibility, and would allow their paper to have immediate impact for developers wanting to use their code or researchers wanting to build off of their work.

Weaknesses: (1) The experimental results did not quite convince me that their proposed approach outperforms prior work. The authors claim in the abstract that PAM is a current state-of-the-art technique, but it is from 1990 and there have been several improvements since then (in fact, all of the algorithms that the authors compared to except for FastPAM1 are at least a decade old). Specifically, the authors compared to FastPAM1 in [1], but in most of the experiments in [1], FastPAM1 was outperformed by FastPAM2, FastCLARA, and FastCLARANS. Also, how does this algorithm compare to [2], which won a best paper award at AISTATS 2017 and runs in O(n^{3/2}) time? (2) (minor) It seems the runtime of their algorithm is actually O(nk log n), and the way the title and intro are written seem like false advertising, because most clustering research does not treat k as a constant. The authors do clarify this later in the intro, when they say that they treat k as a constant for the remainder of the paper. (3) (minor) Can theorems 1 and 2 be proven for an arbitrary delta < 1/n^2? And then the probability of success and the #_distance_computations would be in terms of delta. Small comments: - The phrase “theoretically prove” is redundant (in the abstract and intro) - Eq. 4, line 133, and again in other places. Shouldn’t the carat be min(., .) ? - Pick a variable to represent “batchsize” in theorem 2? Even though I would like to see these weaknesses addressed, I think the pros marginally outweigh the cons. [1] Erich Schubert, Peter Rousseeuw. Faster k-medoids clustering: improving the PAM, CLARA, and CLARANS Algorithms. [2] James Newling, Francois Fleuret. A Sub-Quadratic Exact Medoid Algorithm. 2017.

Correctness: The claims and empirical methodology are correct.

Clarity: The paper is well written.

Relation to Prior Work: It is clearly discussed how this work differs from previous contributions.

Reproducibility: Yes

Additional Feedback:

[Author Response · NeurIPS 2020]

We thank the reviewers for their time, insightful comments, and feedback. We provide a point-by-point response below.

**Reviewer 1:** We agree with Reviewer 1 and will clarify our main message: that Bandit-PAM can achieve the same loss
as state-of-the-art but in much less time. We agree that wall-clock time comparisons are important for this message and,
with that in mind, we ran new experiments comparing Bandit-PAM with FastPAM and FastPAM1, and will add a figure
demonstrating wall-clock time scaling with dataset size to the final paper. For example, in the additional experiments,
Bandit-PAM is 2.7x (3.4x) faster than FastPAM (FastPAM1, respectively) on a subset of MNIST of size $N = 65,000$
and 3.2x (4.0x) faster than FastPAM (FastPAM1, respectively) on the full dataset. We would like to clarify that we
decided to focus on the number of distance evaluations because wall-clock time depends on implementation details
and is not a perfect proxy for algorithmic complexity. We would also like to explain that FastPAM is a faster variant
of FastPAM1 and produces a similar loss, but is not guaranteed to return the same output as PAM/FastPAM1/Bandit-
PAM. Therefore, we think it is more appropriate to compare Bandit-PAM to FastPAM1 conceptually, but to FastPAM
empirically for complexity. We will explain this and also add FastPAM1 results to all experimental results in the paper.

The reason why we stated that Bandit-PAM can be viewed as a batched version of the UCB algorithm is that it uses
upper and lower confidence bounds to discard points. In the final paper, we will be more specific about the relationship
to UCB and also mention the connection with the successive elimination algorithm by Even-Dar et al. Regarding the
question about confidence intervals in the plots, they are the standard error of the empirical average over 10 repetitions,
and are indeed small. Regarding Reviewer 1's other comments, we will include more details regarding the reduction
from $\sum_i \Delta_i^{-2} \log n$ to $O(n \log n)$, clarify that Theorem 1 is a result of $(k + T)$ applications of Theorem 2, and make
the $O(d)$ dependence explicit. We will also define $\sigma_x$ before its first use and enlarge the legends of the plots.

**Reviewer 2:** We would like to clarify what we see as the novelty of our work. We emphasize that the general $k$-medoids
problem is NP-hard, while 1-medoid is not, so the technique in Bagaria et al. 2018 cannot be directly applied when
moving from 1-medoid to $k$-medoids. As such, we present a heuristic generalization of the approach in Bagaria et
al. 2018 in which we track PAM's optimization path and show that this new problem can be efficiently solved as a
sequence of bandit problems. This insight requires different objects to be formulated as bandit "arms" in the BUILD
and SWAP steps, which we consider nontrivial. In contrast, the 1-medoid problem presented in Bagaria et al. 2018 is
exactly solvable in $O(n^2)$ time, requires no heuristic solution, and the technique presented therein is a single bandit
problem. Fundamentally, our work has achieved an $O(n^2)$ to $O(n \log n)$ reduction in a classical clustering problem.
Therefore, we would respectfully argue that our paper is beyond a simple adaptation of prior work and our contribution
is not small as suggested by Reviewer 2.

In addition to providing an efficient $k$-medoids algorithm, we provide an optimized C++ implementation alongside
our paper. We anticipate this implementation will enable clustering large datasets with hard-to-compute distances
using $k$-medoids, such as in the MOOC example presented in the paper. We think this is a valuable contribution to the
applied ML community and can be used in various applications where $k$-medoids algorithms are currently used such as
healthcare, education, operations research, etc.

Reviewer 2 observed that our theoretical result, as stated, would be vacuous if $T$ is large. We would like to offer the
following counterpoints. First, we would like to clarify that, as long as $T$ is poly(n), one could easily modify the proof
to get the same bound with slightly worse constants, by taking the hyperparameter $\delta$ also to be 1/poly(n). Second, we
would also like to note that $T$ has been empirically observed to be $O(k)$ (e.g., Figure 3a in Schubert et al. 2019); indeed,
$T < 2k$ in all our experiments, including in additional experiments we ran for $k = 30$ and $k = 50$. Third, in additional
experiments, we observed that Bandit-PAM is consistent with PAM in 599 out of 600 calls to Algorithm 1 (BUILD or
SWAP steps). We will add a discussion about this to the final paper.

Following Reviewer 2's suggestion, we ran experiments on the total number of unique pairwise distances used. When
$N = 1,000$, all $10^6$ pairwise distances were used and, when $N = 70,000$, 8.7% of distances were used. We will add
these statistics in the final paper and discuss intelligently caching distance computations in future work. We agree that
[Aghaee 2016] is related to lowering the number of unique distance computations (and designing the cache); we will
discuss it and cite it in our final paper.

**Reviewer 5:** Despite being from 1990, PAM is actually considered state-of-the-art in clustering quality (although not
in runtime) according to Schubert et al. 2019. FastPAM produces a similar loss as PAM/FastPAM1/FastPAM2 and
is the fastest, which makes it the focal point for our comparisons. Other algorithms, including CLARA, CLARANS,
FastCLARA, and FastCLARANS produce empirically worse clustering results than PAM (see Figure 5 from Schubert
et al. 2019 and Figure 1a in our work). trimed, from Newling et al. 2017, scales exponentially in the dimension $d$ and
hence is omitted from comparison, as Bandit-PAM scales as $O(d)$. We will clarify these in the final paper and cite
Newling et al. 2017. In addition, we will also make the dependency on $k$ explicit. Reviewer 5 also astutely observes
that Theorems 1 and 2 can be written in terms of general $\delta < n^{-2}$, where $\delta = n^{-3}$ was chosen for convenience. We
will include the result for general $\delta < n^{-2}$ in the paper.

[Meta-Review · NeurIPS 2020]

Two of the three reviewers support acceptance of this paper, and the reviewers all agree that efficient k-median clustering is an important and useful problem. Reviewers 1 and 5 agree that the algorithm is practical and provides significant speedups over state of the art k-median algorithms. A common concern of the reviewers is that the techniques used to obtain the speedups are not too novel given the work of Bagaria et al. (2018) on efficiently computing the median of a set of points. During the discussion period, the reviewers agreed that the results of the current paper do not immediately follow from the prior work, but that the main strength of the paper is the practicality of the algorithm. Since the proposed algorithm is practical, provides a significantly faster algorithm for PAM clustering, and has a very high quality code submission, I recommend accept. The authors should add the experiments and clarifying comments discussed in the reviews and the author response to the camera ready version of the paper.